# Naldemedine is associated with earlier defecation in critically ill patients with opioid-induced constipation: A retrospective, single-center cohort study

Seiya Nishiyama[1], Shigehiko Uchino[1], Yusuke Sasabuchi[2], Tomoyuki Masuyama[1‡]*, Alan Kawarai Lefor[3‡], Masamitsu Sanui[1‡]

1 Department of Anesthesiology and Critical Care Medicine, Jichi Medical University, Saitama Medical Center, Omiya, Saitama, Japan, 2 Data Science Center, Jichi Medical University, Shimotsuke, Tochigi, Japan, 3 Department of Surgery, Jichi Medical University, Shimotsuke, Tochigi, Japan

☯ These authors contributed equally to this work.
‡ TM, AKL and MS also contributed equally to this work.
* tomomasuyama@jichi.ac.jp

## Abstract

### Introduction

There are few reports describing the association of naldemedine with defecation in critically ill patients with opioid-induced constipation. The purpose of this study was to determine whether naldemedine is associated with earlier defecation in critically ill patients with opioid-induced constipation.

### Methods

In this retrospective cohort study, patients admitted to the Intensive Care Unit (ICU) without defecation for 48 hours while receiving opioids were eligible for enrollment. The primary endpoint was the time of the first defecation within 96 hours after inclusion. Secondary endpoints included presence of diarrhea, duration of mechanical ventilation, ICU length of stay, ICU mortality, and in-hospital mortality. The Cox proportional hazard regression analysis with time-dependent covariates was used to evaluate the association naldemedine with earlier defecation.

### Results

A total of 875 patients were enrolled and were divided into 63 patients treated with naldemedine and 812 patients not treated. Defecation was observed in 58.7% of the naldemedine group and 48.8% of the no-naldemedine group during the study (p = 0.150). The naldemedine group had statistically significantly prolonged duration of mechanical ventilation (8.7 days vs 5.5 days, p < 0.001) and ICU length of stay (11.8 days vs 9.2 days, p = 0.001) compared to the no-naldemedine group. However, the administration of naldemedine was significantly associated with earlier defecation [hazard ratio:2.53; 95% confidence interval: 1.71–3.75, p < 0.001].

**Data Availability Statement:** All relevant data are within the manuscript and its Supporting Information files.

**Funding:** The authors received no specific funding for this work.

**Competing interests:** The authors have declared that no competing interests exist.

## Conclusion

The present study shows that naldemedine is associated with earlier defecation in critically ill patients with opioid-induced constipation.

## Introduction

Bowel dysfunction is an important problem in patients in the intensive care unit (ICU), where approximately 70% of critically ill patients suffer from constipation [1]. Bowel dysfunction in critically ill patients is associated with adverse outcomes including gastroesophageal reflux, delayed gastric emptying leading to increased aspiration, decreased enteral feeding, delayed ICU discharge, and increased mortality [2–4]. Among the three classes of opioid receptors in the gastrointestinal tract (μ, κ, and δ), the κ- and δ- receptors are expressed primarily in the stomach and proximal colon, while the μ-receptors are widely expressed throughout the gastrointestinal tract. Activation of μ-opioid receptors by exogenous opioids, widely used in the ICU, decreases gastric, small and large intestinal motility, decreases gastrointestinal, biliary and pancreatic secretions, increases water absorption from enteric contents, and results in contraction of the pyloric and anal sphincters [5]. As a result, opioid-induced constipation (OIC) is the most common manifestation of opioid-induced bowel dysfunction [6], occurring in 15–80% of patients receiving opioids [7–12].

Peripherally acting μ-opioid receptor antagonists (PAMORAs) were developed to antagonize only peripheral μ-opioid receptors because of their limited ability to cross the blood brain barrier to avoid impairing the analgesic effect [13]. There are now three PAMORAs (methylnaltrexone, naldemedine, naloxegol) commercially available [12]. Naldemedine has been reported to be effective for the treatment of OIC in chronic non-cancer pain (COMPOSE-1 and COMPOSE-2 trials) [14] and in patients with cancer (COMPOSE-4, COMPOSE-5 trials) [15]. In these studies, naldemedine 0.2 mg/day significantly increased the frequency of spontaneous bowel movements compared to placebo in patients receiving long-term opioids.

However, few studies have demonstrated that PAMORAs are associated with defecation in critically ill patients with OIC. The MOTION trial investigated the effects of methylnaltrexone in adult ICU patients on mechanical ventilation while receiving opioids [16]. There was no difference in time to rescue-free defecation between methylnaltrexone and placebo in patients constipated for more than 48 hours, with each institutional laxative protocol.

Naldemedine has strong binding affinity and antagonistic activity at μ-, δ-, and κ-opioid receptors differing from that of methylnaltrexone [17]. However, to the best of our knowledge, its association with defecation has not been evaluated in critically ill patients with constipation. We hypothesized that naldemedine would reduce time to defecation in critically ill patients with OIC. To investigate the association of naldemedine with time to defecation, a retrospective study of critically ill patients receiving opioids in a single ICU was conducted.

## Material and methods

### Study design

This was a retrospective cohort study that evaluated the association naldemedine with earlier defecation in patients with constipation who were treated with opioids in the ICU at Jichi Medical University Saitama Medical Center. This study was conducted according to the ethical principles outlined in the Declaration of Helsinki and was approved by the Institutional

Review Board of Jichi Medical University Saitama Medical Center (request number: S21-185). The consent and ethics approval for each patient by the research team were waived because of the use of anonymized existing data.

### Study population

Eligibility criteria included all patients admitted to the ICU from August 1, 2017, to September 30, 2021. Patients were included if they were constipated (defined as absence of any stool evacuation) for 48 hours while receiving opioids [16]. For each enrolled patient, only the initial episode of constipation was included in the analysis. Patients were excluded if they were under 18 years old, stayed in the ICU for less than 48 hours, had ileostomy or colostomy, or had missing data.

### Exposure

Patients who received at least one dose of naldemedine (Symproic®, Shionogi Inc.) during the observation period were defined as the 'naldemedine group'. Naldemedine was administered orally or via gastric tube at 0.2 mg once daily based on the package insert. Patients who did not receive naldemedine at the time of inclusion or during the observation period were defined as the 'no-naldemedine group'. Rescue laxatives were defined as a new laxative administration or dosage increase of previous laxatives after inclusion. Based on bowel movement records, rescue laxatives were prescribed at the discretion of the critical care physicians.

### Collected variables

Data were retrospectively collected from the hospital's electronic patient management system for critical care (ACSYS-Ki™, PHILIPS Japan, Tokyo, Japan). The access to database to obtain the data used in this study was on May 11, 2022. The authors did not have access to information that could identify individual participants after data collection. Patient's baseline characteristics on admission, including demographic data and comorbidities were collected. Laboratory data such as serum bilirubin, creatinine, and blood gas analysis needed for $PaO_2/FiO_2$ ratio were collected at the time of inclusion or within 24 hours prior. Use of medications such as antibiotics, sedatives, vasoactive drugs, muscle relaxants, and laxatives, as well as enteral nutrition, and the Richmond agitation sedation score (RASS) were evaluated at the time of inclusion. The date and time of administration of naldemedine, administration of rescue laxatives, and the first defecation after inclusion were obtained from the electronic medical record.

### Outcome measures

The primary endpoint was defecation within 96 hours. The time of defecation was defined as the time of the first bowel movement of any consistency. Secondary endpoints were diarrhea within 96 hours, mechanical ventilation period during ICU stay, ICU length of stay, ICU mortality, and in-hospital mortality.

### Statistical methods

Data were expressed as medians (25–75% interquartile range) or means (standard deviation), where appropriate. The Fisher's exact test and Wilcoxon's rank sum test were used to compare the naldemedine group and the no-naldemedine group. The 'Defecation group' and the 'No-defecation group', defined as defecation within 96 hours or not, were also compared using Fisher's exact test and Wilcoxon's rank-sum test.

The Cox proportional hazard regression analysis with time-dependent covariates was used to evaluate the association naldemedine with earlier defecation. The administration of naldemedine or rescue laxatives from time of inclusion until outcome occurrence was considered as a time-dependent variable. The observation period was 96 hours from inclusion or ICU discharge, whichever came first: Patients who left the ICU without defecation within the observation period or patients who reached 96 hours without defecation were censored.

Several subgroup analyses were also conducted to calculate hazard ratios for the time to defecation from inclusion for naldemedine administration. Specific subgroups included age ($\leqq$60 years, >60 years to <75 years, $\geqq$75 years), body mass index (<30 kg/m$^2$ or $\geq$30 kg/m$^2$), sex (male or female), PaO$_2$/FiO$_2$ ratio (>200 mmHg, >100 mmHg to $\leqq$200 mmHg, <100 mmHg), main disease category (cardiovascular, respiratory, neurological, abdominal, others), the prior use of laxatives (yes or no), maintenance hemodialysis (yes or no), mechanical ventilation (yes or no), use of antibiotics (yes or no), receiving other sedatives (yes or no), receiving vasoactive drugs (yes or no), and use of enteral nutrition (yes or no). The calculated hazard ratios were shown graphically in a forest plot.

In all analyses, two-sided tests were used to determine the statistical significance at a 0.05 level. Statistical analyses were performed using R statistical language version 4.0.5 (The R Foundation, Vienna, Austria) and EZR version 1.54 [18]. Data analysis was conducted from May 12 to July 10, 2022.

## Results

Of 6,804 patients screened, 4,552 met the inclusion criteria of constipation for 48 hours while receiving opioids. The following patients were excluded; 44 patients under 18 years old, 3,461 patients who stayed in the ICU for less than 48 hours, 117 patients with an ileostomy or colostomy, and 55 patients with missing data. Finally, 875 patients were enrolled for analysis and were grouped into 63 patients treated with naldemedine and 812 patients not treated with naldemedine during the observation period, respectively (Fig 1).

Patient demographics and baseline characteristics are shown in Table 1. There were no significant differences in age, gender, or APACHE II score between the groups. The naldemedine group had a significantly higher BMI than the no-naldemedine group (24.2 kg/ vs. 22.9 kg/, p = 0.047). The naldemedine group had more ICU admissions for medical illnesses (60.3% vs. 44.3%), while the no-naldemedine group had more post-operative ICU admissions (55.7% vs. 39.7%). Notably, there were more respiratory diseases in the naldemedine group and significantly fewer gastrointestinal diseases than in the no-naldemedine group. There were no patients receiving maintenance dialysis in the naldemedine group. Most patients underwent mechanical ventilation (92.3% of the total cohort). Significantly more patients in the naldemedine group were treated with sedatives (73.0% vs. 44.8%, p <0.001) and muscle relaxants (12.7% vs. 2.7%, p = 0.001) than patients in the no-naldemedine group. In addition, more patients in the naldemedine group received antibiotics therapy than in the no-naldemedine group (66.7% vs. 36.7%). Almost all the opioids administered were fentanyl, while morphine was used in few patients (5 patients, 0.6% of overall subjects), and some treated with both morphine and fentanyl. There were no significant differences between the two groups regarding the numbers of patients receiving enteral nutrition, or any laxative at inclusion. Patient characteristics stratified by defecation status are shown in S1 Table.

Defecation was observed in 58.7% of the naldemedine group and 48.8% of the no-naldemedine group during the observation period (Table 2). In adddition, 41.3% of the naldemedine group and 16.7% of the no-naldemedine group received rescue laxatives during the observation period (p < 0.001). Rescue laxatives received during the study are shown in S2 Table.

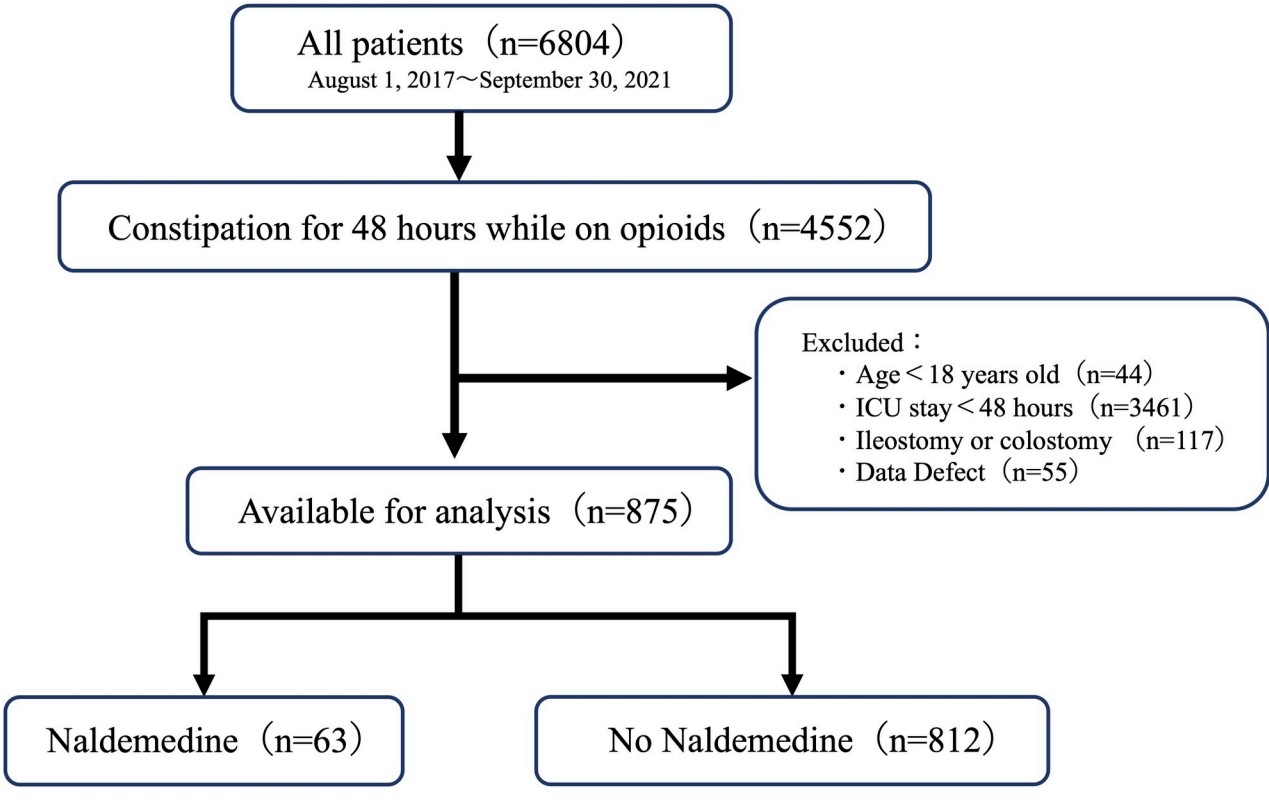

**Fig 1. Flow diagram of patient inclusion and exclusion in the present study.** *ICU* Intensive Care Unit.

Defecation without rescue laxatives was observed in 33.3% of the naldemedine group and 41.5% of the no-naldemedine group (p = 0.232). There were no differences in the prevalence of diarrhea comparing the naldemedine and no-naldemedine groups (11.1% vs. 12.7%, p = 0.845). The naldemedine group had significantly prolonged mechanical ventilation duration (8.7 days vs. 5.5 days, p < 0.001) and ICU length of stay (11.8 days vs. 9.2 days, p = 0.001) compared to the no-naldemedine group. There were no differences in hospital or ICU mortality between the two groups. Multivariable Cox proportional hazard regression analysis with time-dependent covariate showed that naldemedine administration was significantly associated with earlier defecation [hazard ratio (HR):2.53; 95% confidence interval (95% CI): 1.71–3.75, p < 0.001] (Table 3). The point estimates in each subgroup were mostly in the direction of the effect of naldemedine administration (Fig 2).

Hazard ratios (black squares), 95% CIs (horizontal lines), p-values are for the interaction between the treatment effect and subgroup variables. The point estimates in each subgroup were mostly in the direction of the effect of naldemedine administration.

## Discussion

### Key findings

This study was conducted to investigate whether naldemedine is associated with earlier defecation in critically ill patients who have not defecated for 48 hours while receiving opioid therapy (more than 99% of opioid doses administered were fentanyl). Despite some differences in patient characteristics, the present study showed that naldemedine is significantly associated with earlier defecation after adjusting for differences [HR:2.53, p < 0.001]. The results were

**Table 1. Patient characteristics stratified by naldemedine use.**

| | All | Naldemedine | No Naldemedine | P value |
|---|---|---|---|---|
| Number of patients | 875 | 63 | 812 | |
| Age [median (IQR)] | 69 [56–76] | 60 [46–78] | 69 [57–76] | 0.146 |
| Males [n (%)] | 596 (68.1) | 47 (74.6) | 549 (67.6) | 0.326 |
| BMI (kg/) [median (IQR)] | 23.0 [20.1–26.0] | 24.2 [21.1–26.6] | 22.9 [20.0–25.9] | 0.047 |
| Maintenance hemodialysis | 51 (5.8) | 0 (0.0) | 51 (6.3) | 0.044 |
| Reason for ICU admission [n (%)] | | | | 0.018 |
| Medical (non-operative) | 398 (45.5) | 38 (60.3) | 360 (44.3) | |
| Surgical—elective (operative) | 211 (24.1) | 8 (12.7) | 203 (25.0) | |
| Surgical—emergency (operative) | 266 (30.4) | 17 (27.0) | 249 (30.7) | |
| Main disease category [n (%)] | | | | 0.001 |
| Cardiovascular | 395 (45.1) | 28 (44.4) | 367 (45.2) | |
| Respiratory | 207 (23.7) | 27 (42.9) | 180 (22.2) | |
| Neurological | 92 (10.5) | 5 (7.9) | 87 (10.7) | |
| Abdominal | 60 (6.9) | 1 (1.6) | 59 (7.3) | |
| Others | 121 (13.8) | 2 (3.2) | 119 (14.7) | |
| APACHE II score [median (IQR)] | 20 [15–25] | 19 [13–25] | 20 [15–25] | 0.110 |
| Type of opioid [n (%)] | | | | |
| Fentanyl | 871 (99.5) | 62 (98.4) | 809 (99.6) | 0.259 |
| Morphine | 5 (0.6) | 1 (1.6) | 4 (0.5) | 0.312 |
| Mechanical ventilation [n (%)] | 808 (92.3) | 62 (98.4) | 746 (91.9) | 0.080 |
| Receiving other sedatives [n (%)] | 410 (46.9) | 46 (73.0) | 364 (44.8) | <0.001 |
| Receiving vasoactive drugs [n (%)] | 258 (29.5) | 19 (30.2) | 239 (29.4) | 0.887 |
| Receiving muscle relaxants [n (%)] | 30 (3.4) | 8 (12.7) | 22 (2.7) | 0.001 |
| PaO2/FiO2 (mmHg) [median (IQR)] | 240 [175–308] | 195 [154–248] | 245 [179–311] | <0.001 |
| Creatinine (mg/dL) [median (IQR)] | 0.97 [0.66–1.78] | 0.76 [0.62–1.35] | 0.97 [0.66–1.81] | 0.037 |
| Bilirubin (mg/dL) [median (IQR)] | 0.73 [0.46–1.25] | 0.73 [0.41–1.31] | 0.73 [0.46–1.24] | 0.579 |
| RASS [median (IQR)] | -1 [−3− −1] | -2 [−4− −1] | -1 [−3− −1] | 0.015 |
| Antibiotics use [n (%)] | 340 (38.9) | 42 (66.7) | 298 (36.7) | <0.001 |
| Enteral nutrition use [n (%)] | 431 (49.3) | 36 (57.1) | 395 (48.6) | 0.239 |
| Laxative at inclusion [n (%)] | 37 (4.2) | 2 (3.2) | 35 (4.3) | 1.000 |
| Metoclopramide use [n (%)] | 42 (4.8) | 5 (7.9) | 37 (4.6) | 0.218 |

BMI: body mass index, GI: Gastrointestinal, APACHE II: acute physiology and chronic health evaluation II, RASS: Richmond agitation sedation score.

also similar in most subgroup analyses. There was no difference in the occurrence of diarrhea between the naldemedine and the no-naldemedine groups (11.1% vs 12.7%).

## Relationship with previous studies

Previous studies have shown that naldemedine reduces the incidence of OIC in patients with non-cancer chronic pain (COMPOSE-1, and -2) and patients with cancer (COMPOSE-4, and -5) [14, 15]. In these studies, naldemedine 0.2 mg/day or placebo was administered during the 2 to 12 week study period, and the primary outcome was based on the proportion of responders defined as spontaneous bowel movements ≥3 per week and ≥1 increase per week from baseline. Compared to those studies, the maximum observation period of 96 hours in our study was relatively short. In all previous studies, the proportion of responders was significantly higher in the naldemedine group than in the placebo group (COMPOSE-1: 47.6% vs 34.6, COMPOSE-2: 52.5% vs 33.6, COMPOSE-4: 71.1% vs 34.4%). In the present study, the

**Table 2. Patient outcomes.**

|  | All | Naldemedine | No Naldemedine | P value |
|---|---|---|---|---|
| Number of patients | 875 | 63 | 812 |  |
| Rescue laxative [n (%)] | 162 (18.5) | 26 (41.3) | 136 (16.7) | <0.001 |
| Defecation [n (%)] | 433 (49.5) | 37 (58.7) | 396 (48.8) | 0.150 |
| Defecation without rescue laxatives [n (%)] | 358 (40.9) | 21 (33.3) | 337 (41.5) | 0.232 |
| Diarrhea [n (%)] | 110 (12.6) | 7 (11.1) | 103 (12.7) | 0.845 |
| Ventilation period [days (IQR)] | 5.7 [2.9–12.4] | 8.7 [5.5–16.4] | 5.5 [2.8–12.2] | <0.001 |
| ICU discharge on MV [n (%)] | 227 (25.9) | 15 (23.8) | 212 (26.1) | 0.767 |
| ICU death while on MV [n (%)] | 108 (12.3) | 7 (11.1) | 101 (12.4) | 1.000 |
| ICU length of stay [days (IQR)] | 9.6 [5.8–16.3] | 11.8 [8.9–18.5] | 9.2 [5.7–16.1] | 0.001 |
| ICU mortality [n (%)] | 114 (13.0) | 8 (12.7) | 106 (13.1) | 1.000 |
| In-hospital mortality [n (%)] | 188 (21.5) | 12 (19.0) | 176 (21.7) | 0.750 |

MV: Mechanical ventilation, ICU: Intensive Care Unit, IQR: Interquartile range

**Table 3. Cox proportional hazard regression analysis with time-dependent covariate for defecation.**

| Variable | HR | 95% CI | P Value |
|---|---|---|---|
| Age (year) | 1.01 | 0.99–1.02 | 0.063 |
| Males | 1.10 | 0.88–1.38 | 0.385 |
| BMI (kg/) | 0.97 | 0.94–1.00 | 0.027 |
| Maintenance hemodialysis | 0.90 | 0.54–1.50 | 0.688 |
| Reason for ICU admission |  |  |  |
| Medical (non-operative) |  | Reference |  |
| Surgical—elective (operative) | 0.77 | 0.56–1.05 | 0.092 |
| Surgical—emergency (operative) | 0.84 | 0.64–1.11 | 0.218 |
| Main disease category |  |  |  |
| Cardiovascular |  | Reference |  |
| Pulmonary | 1.12 | 0.82–1.53 | 0.470 |
| Neurological | 0.93 | 0.62–1.37 | 0.701 |
| Abdominal | 1.26 | 0.80–1.99 | 0.314 |
| Others | 1.21 | 0.85–1.73 | 0.275 |
| APACHE II score | 1.02 | 1.00–1.04 | 0.007 |
| Mechanical ventilation | 0.79 | 0.51–1.22 | 0.281 |
| Other sedatives | 0.74 | 0.60–0.92 | 0.006 |
| Vasoactive drugs | 0.78 | 0.60–1.00 | 0.049 |
| Muscle relaxants | 1.27 | 0.78–2.05 | 0.328 |
| $PaO_2/FiO_2$ (mmHg) | 1.00 | 0.99–1.01 | 0.847 |
| Creatinine (mg/dL) | 1.09 | 1.00–1.19 | 0.038 |
| Bilirubin (mg/dL) | 1.02 | 0.96–1.07 | 0.528 |
| RASS | 1.09 | 1.02–1.17 | 0.010 |
| Antibiotics use | 0.95 | 0.76–1.18 | 0.628 |
| Enteral nutrition use | 1.28 | 1.01–1.62 | 0.035 |
| Rescue laxative | 1.65 | 1.25–2.17 | <0.001 |
| Naldemedine | 2.53 | 1.71–3.75 | <0.001 |

BMI: body mass index, GI: Gastrointestinal, APACHE II: acute physiology and chronic health evaluation II, RASS: Richmond agitation sedation score, HR: hazard ratio, CI: confidence interval. ICU: Intensive Care Unit

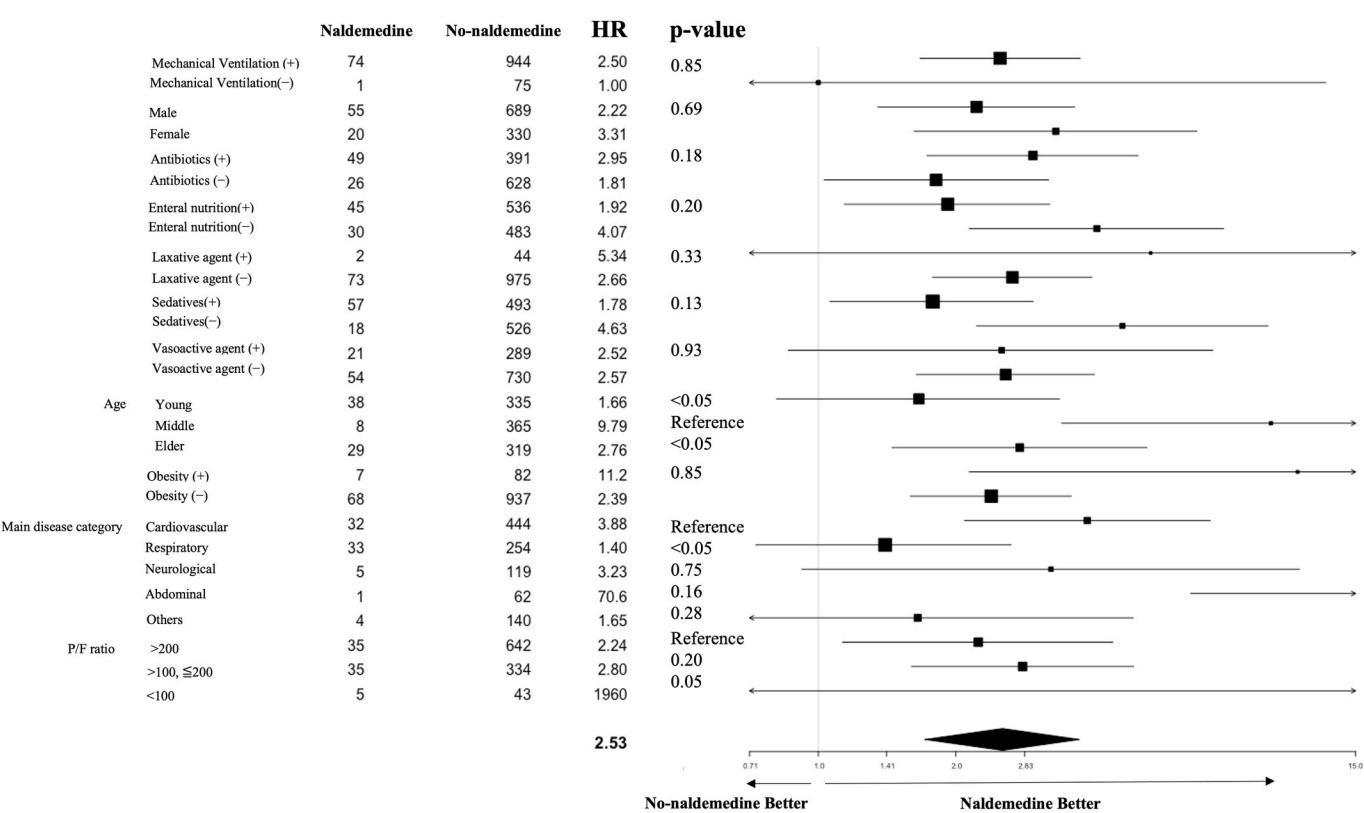

**Fig 2. Forest plot showing the results of subgroup analysis for the primary outcome.**

proportion of patients who had defecation within the observation period tended to be higher in the naldemedine group than in the no-naldemedine group (58.7% vs. 48.8%).

The MOTION trial was conducted in critically ill patients with OIC to evaluate the efficacy of methylnaltrexone (a type of PAMORA) but did not show an effect on time to defecation compared to placebo [16]. The present study also included critically ill patients in the ICU, and showed an association of naldemedine with earlier defecation. Although several systematic reviews and meta-analyses have suggested superiority or inferiority for each PAMORA with rescue-free defecation and rescue laxative reduction as outcomes, no head-to-head comparison of PAMORAs has been conducted to date [19]. Although methylnaltrexone and naldemedine belong to the group of PAMORAs, they have different structures to prevent passage through the blood brain barrier [6]. Furthermore, methylnaltrexone antagonizes peripheral μ-receptors, while naldemedine antagonizes peripheral μ-, δ- and κ-receptors [6]. Such differences between naldemedine and methylnaltrexone may have affected the observed differences in study results.

Previous studies have shown that diarrhea was the most frequent adverse event in patients treated with naldemedine (7 ∼ 19.6%) and was also more frequent compared to patients treated with placebo (2 ∼ 7.3%) [14, 15]. In the present study, the incidence of diarrhea in the naldemedine group was 11.1%, similar to the previous report. However, there was no difference in the incidence of diarrhea between the two groups in the present study. It is known that with longer duration of opioid therapy prior to naldemedine administration, there is a greater incidence of naldemedine-induced diarrhea. Okamoto and colleagues reported that opioid therapy for more than 17 days prior to naldemedine administration (odds ratio [OR] = 7.539,

P = 0.016) [20] was an independent risk factor for the development of naldemedine-induced diarrhea. Since patients in previous studies received opioid therapy for 2–4 weeks or longer [14, 15], the incidence of naldemedine-associated diarrhea may have been less frequent, in the population evaluated in the present study (most patients were treated with opioids after admission to the ICU).

## Significance and implications

Opioids are widely used in the ICU, and OIC is the most common manifestations of opioid-induced bowel dysfunction and cause of constipation in ICU patients. OIC is often difficult to manage, but PAMORAs such as naldemedine can potentially improve OIC in the ICU patients. There have been some efforts to directly antagonize intestinal opioid receptors with specific pharmacologic agents. Enteral naloxone administration was used for this approach and was shown to reduce gastric tube reflux volume and reduce pneumonia, but there was no difference in time to first bowel movement compared to placebo [21]. Opioid withdrawal is often the problem and can result in exacerbation of pain [22]. Naldemedine, acting only on peripheral μ opioid receptors, is known to significantly reduce deterioration in quality of defecation in patients with and without cancer receiving opioids compared to magnesium oxide, and to reduce gastrointestinal side effects such as nausea [23]. Naldemedine also has been shown to improve OIC in patients without opioid withdrawal [24] and may have an important role in the treatment of constipation in critically ill patients requiring strict pain management. The results of the present study suggest for the first time that naldemedine may be associated with earlier defecation for patients with OIC in the ICU. Future prospective studies should compare naldemedine with other agents to treat patients in the ICU to verify the effects of naldemedine on bowel movements.

## Strengths and limitations

There have been few studies evaluating the association of PAMORAs with time to defecation in critically ill patients with OIC. To the best of our knowledge, this is the first study to demonstrate that naldemedine is associated with decreased time to defecation in critically ill patients receiving opioids. Diarrhea, which has been reported as a major adverse event of naldemedine, was not associated with naldemedine administration in the present study. Furthermore, although there are multiple factors affecting defecation in patients in the ICU (such as administration of antimicrobial agents, surgery, enteral nutrition, mechanical ventilation, vasoactive drugs, sedatives, and disease severity) [1, 25–29], the sample size in the present study (433 patients with defecation during the observation period) was large enough to include all factors associated with defecation in previous studies and in clinical practice as explanatory variables [30].

The present study has acknowledged limitations. First, because of the retrospective nature of this study, we could not completely exclude the confounding factors. Second, secondary outcomes, including duration of mechanical ventilation and ICU length of stay, were significantly longer in the naldemedine group. This may have been influenced by the higher number of respiratory disease patients in the naldemedine group, including novel coronavirus infections (COVID-19). More patients in the naldemedine group were receiving mechanical ventilation, showed lower $PaO_2$ / $FiO_2$ ratios, and received other sedatives and muscle relaxants compared to the no-naldemedine group. Finally, more patients in the naldemedine group received rescue laxatives than patients in the no-naldemedine group. An adjustment for administration of rescue laxatives as an explanatory variable was made in multivariate analysis and the resulting data showed a significant association between naldemedine and earlier defecation.

## Conclusions

The results of the present study show that naldemedine, one of the PAMORAs, is associated with earlier defecation in critically ill patients with OIC. Subset analyses also showed similar results. Naldemedine may be a treatment option for constipation in patients treated with opioids in the ICU.

## Supporting information

**S1 Table. Patient characteristics stratified by defecation status.**
(DOCX)

**S2 Table. Rescue laxatives received during the study.**
(DOCX)

**S1 Data. Data set to reach the conclusion.**
(XLSX)

**S1 File. Explanations of variables in the data set.**
(DOCX)

## Author Contributions

**Conceptualization:** Seiya Nishiyama.

**Data curation:** Seiya Nishiyama, Shigehiko Uchino.

**Formal analysis:** Seiya Nishiyama, Yusuke Sasabuchi.

**Methodology:** Seiya Nishiyama, Shigehiko Uchino, Yusuke Sasabuchi, Tomoyuki Masuyama, Alan Kawarai Lefor, Masamitsu Sanui.

**Writing – original draft:** Seiya Nishiyama.

**Writing – review & editing:** Seiya Nishiyama, Shigehiko Uchino, Yusuke Sasabuchi, Tomoyuki Masuyama, Alan Kawarai Lefor, Masamitsu Sanui.

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
