## [Decision Letter · Decision Letter 0]

13 Sep 2023

PONE-D-23-14424Naldemedine is associated with earlier defecation in critically ill patients with opioid-induced constipation: A retrospective, single-center cohort study.PLOS ONE

Dear Dr. Masuyama,

Thank you for submitting your manuscript to PLOS ONE. After careful consideration, we feel that it has merit but does not fully meet PLOS ONE’s publication criteria as it currently stands. Therefore, we invite you to submit a revised version of the manuscript that addresses the points raised during the review process.

We look forward to receiving your revised manuscript.

Kind regards,

Guilherme Antonio Moreira de Barros, M.D., M.Sc., Ph.D

Academic Editor

PLOS ONE

“NO”

Additional Editor Comments:

Dear Authors,

Thank you for submitting your work to the Plos One Journal. We appreciate your efforts and would like to remember you some important points of review process (these concerns may or may not be pointed as issues on your submission, or may not be related to the type of the manuscript you presented):

1) We advise the uso of the Enhancing the Quality and Transparency of Health Research (EQUATOR) reporting guidelines and checklists in order to improve the comprehension and aid critical appraisal of your work. Please select the appropriate reporting guideline for your manuscript at www.equator-network.org/toolkits and follow the instructions provided by the consolidated statement that better suit your study type.

2) Good quality journals embrace a high ethical standard. Informed consent and Ethics Committee or Institutional Board Review approvals must be presented and stated in the manuscript text for studies or reports involving human and animal experiments. Anonymity of participants is imperative, and all efforts must be made to minimize pain and suffering in clinical and animal trials with a clear description of the adopted measures to ensure that in the study methodology.

3) Your references must be in accordance with the journal's specification.

4) Submit your manuscript to a professional language editing service if you are not a native English speaker. Please provide the certificate issued by this service.

5) Submit the methodology and the data analyses to a professional statistician and make sure that all relevant information regarding the descriptive and inferential statistics, as well as the sample size calculation and the randomization or matching process, are accounted for in the statistical description.

6) Please read carefully all reviewers comments, suggestions and questions and include your answers in a response letter highlighting every modifications or alterations that were made in your manuscript. Keep in mind that there must be a proper answer to all issues mentioned in peer review.

7) Plagiarism, even in the form of self-plagiarism, is not accepted by the Brazilian Journal of Anesthesiology. Please manage carefully your citations and quotes and make sure that all efforts were made to avoid improper misuse of ideas or words that are not your own.

8) This is not a guarantee and does not provide assurance that your manuscript will be accepted for publication in the Brazilian Journal of Anesthesiology if you wish to proceed the revision process.

Reviewers' comments:

Reviewer's Responses to Questions

**Comments to the Author**

1. Is the manuscript technically sound, and do the data support the conclusions?

Reviewer #1: Partly

Reviewer #2: Yes

2. Has the statistical analysis been performed appropriately and rigorously? 

Reviewer #1: Yes

Reviewer #2: Yes

3. Have the authors made all data underlying the findings in their manuscript fully available?

Reviewer #1: Yes

Reviewer #2: Yes

4. Is the manuscript presented in an intelligible fashion and written in standard English?

Reviewer #1: Yes

Reviewer #2: Yes

5. Review Comments to the Author

Reviewer #1: 1. The indication and administration details of “rescue laxatives” in this study should be stated.

2. More patients in Naldemedine group received “rescue laxatives” . Is that the reason that more patients defecated in naldemedine group？ How many patients in the naldemedine group defecated without using “rescue laxatives”? While How many patients in the no-naldemedine group defecated without using “rescue laxatives”? Is there any statistical difference？

Reviewer #2: Well written presentation of the study conducted by the authors. Good structured manuscript with detailed methodology and results-presenting section. Clearly noted limitation part. And though the overall control group was relatively small (68 patients) , I feel that the present paper could be published in its present form.

6. PLOS authors have the option to publish the peer review history of their article (what does this mean?). If published, this will include your full peer review and any attached files.

Reviewer #1: No

Reviewer #2: **Yes: **Theodoros Aslanidis

---

## [Author Response · Author response to Decision Letter 0]

23 Oct 2023

EDITOR’S SPECIFIC COMMENTS: 

[Response]

Thank you for your feedback. We have revised the manuscript to align with the style of your journal. We have added author contributions to the first page.

“NO”

[Response]

As suggested, we have incorporated the recommended text on page 18: “Funding: The authors received no specific funding for this work.”, “Competing interests: The authors have declared that no competing interests exist.”

[Response]

Thank you for pointing this out. For the purpose of study reproducibility, we will provide the dataset used in this study. We would like to provide the dataset as "S1 Data" and a variable description of the data as "S1 File”. We have added this information to the Supporting information section on page 24 of the manuscript.

[Response]

Thank you for pointing this out. We have verified that the reference is complete and correct.

REVIEWERS' COMMENTS:

Reviewer #1:

1. The indication and administration details of “rescue laxatives” in this study should be stated.

[Response]

We have added the following statement to the 'Material and Methods’ section (page6, paragraph 1): 'Based on bowel movement records, rescue laxatives were prescribed at the discretion of the critical care physicians.'

2. More patients in Naldemedine group received “rescue laxatives”. Is that the reason that more patients defecated in naldemedine group? How many patients in the naldemedine group defecated without using “rescue laxatives”? While how many patients in the no-naldemedine group defecated without using “rescue laxatives”? Is there any statistical difference?

[Response]

We appreciate these helpful suggestions. After reviewing the data you pointed out, we realized that our previous analysis was partially incorrect. We incorrectly considered patients who were exposed to naldemedine or rescue laxatives after the outcome occurred as 'exposed'. We have provided corrected Data set to reach the conclusion (S1 Data).

Based on your suggestions, we were able to recognize and correct the errors. We performed the analysis again and found that there were 63 patients in ‘Naldemedine group’ and 812 in ‘No naldemedine group'. Similarly, the use of rescue laxative decreased in each group. We analyzed how often patients defecated without rescue laxatives and found no statistically significant difference (p=0.232). We have added this result to Table 2 on page 11 as 'Defecation without rescue laxatives [n (%)]'.

It is important that these analyses are univariate, unadjusted for confounding, and do not affect the results of the main analysis, the Cox proportional hazard regression analysis with time-dependent covariates.

Based on the revised number of participants in each group, we have revised Table 1 (page 10) and Fig. 1. We also revised the Results section (paragraphs 1-3 [pages 8-12]) and Discussion section (paragraphs 1-2 [pages 13-14] and 4 [page 15]), respectively.

Reviewer #2: Well written presentation of the study conducted by the authors. Good structured manuscript with detailed methodology and results-presenting section. Clearly noted limitation part. And though the overall control group was relatively small (68 patients), I feel that the present paper could be published in its present form.

[Response]

We thank the reviewer for these excellent comments. As answered above, the number of participants in ‘Naldemedine group’ decreased slightly. However, this change did not affect the results of the main analysis with time-dependent covariates.

Other minor revisions：

[1] We have changed the 'N' indicating the number of participants in each table to 'n.' This is more suitable for representing the sample size rather than the population.

[2] We have removed the phrase ‘for secondary outcomes’ from the first paragraph of "Statistical methods" section (page 7).

[3] In Table2 (page 11) 'Death while on MV' has been corrected as 'ICU death while on MV'.

[4] The same words ('study') were duplicated in the first paragraph of 'Strengths and limitations' (page 17), so one has been removed.

---

## [Decision Letter · Decision Letter 1]

4 Dec 2023

Naldemedine is associated with earlier defecation in critically ill patients with opioid-induced constipation: A retrospective, single-center cohort study.

PONE-D-23-14424R1

Dear Dr. Masuyama,

We’re pleased to inform you that your manuscript has been judged scientifically suitable for publication and will be formally accepted for publication once it meets all outstanding technical requirements.

Kind regards,

Guilherme Antonio Moreira de Barros, M.D., M.Sc., Ph.D

Academic Editor

PLOS ONE
---

## [Editor Report · Acceptance letter]

7 Dec 2023

PONE-D-23-14424R1 

Naldemedine is associated with earlier defecation in critically ill patients with opioid-induced constipation: A retrospective, single-center cohort study. 

Dear Dr. Masuyama:

I'm pleased to inform you that your manuscript has been deemed suitable for publication in PLOS ONE. Congratulations! Your manuscript is now with our production department. 

Kind regards, 

on behalf of

Dr. Guilherme Antonio Moreira de Barros 

Academic Editor

PLOS ONE